# Steam recovery from flue gas by organosilica membranes for simultaneous harvesting of water and energy

Norihiro Moriyama[1], Akihiro Takeyama[2], Taichi Yamatoko[2], Ken-ichi Sawamura[3], Koji Gonoi[3], Hiroki Nagasawa[1], Masakoto Kanezashi[1] & Toshinori Tsuru [1] ✉

Steam recovery from the spent gases from flues could be a key step in addressing the water shortage issue while additionally benefiting energy saving. Herein, we propose a system that uses organosilica membranes consisting of a developed layered structure to recover steam and latent heat from waste. Proof-of-concept testing is conducted in a running incinerator plant. The proposed system eliminates the need for a water supply while simultaneously recovering latent heat from the waste stream. First, the long-term stability of an organosilica membrane is confirmed over the course of six months on a laboratory-scale under a simulated waste stream. Second, steam recovery is demonstrated in a running waste incinerator plant (bench-scale), which confirms the steady operation of this steam recovery system with a steam recovery rate comparable to that recorded in the laboratory-scale test. Third, process simulation reveals that this system enables water-self-reliance with energy recovery that approximates 70% of waste combustion energy.

Increasing populations will continue to exacerbate water shortage issues. From both industrial and domestic viewpoints, about 4 billion people worldwide currently experience water scarcity for an average of at least one month each year[1]. Industrial water use is almost twice that of domestic use and it will become much larger in the next several decades[2]. In addition, industrial use of energy usually increases $CO_2$ emissions, which are a primary contributor to climate change and all the problems associated with it. One of the worldwide problems worsened by climate change is the decline in the steady supply of fresh water. Therefore, water and energy savings by industries are one key to preventing shortages of water.

Due to its large capacity for generating heat as well as to availability, water and steam are the most commonly used heating mediums in industries such as power plants and chemical plants[3–5]. Most plants combust fossil fuels in the operation of systems that lack energy efficiency. The energy loss is usually due to hot water and/or steam released[6,7], in particular, steam contains a significant amount of latent heat: 2.442 MJ kg$^{-1}$ at 25 °C[8].

In conventional processing, steam is not recovered; it is released from stacks. Here, we describe a system for the recovery of steam when natural gas is used to generate power, as schematically shown in Fig. 1a. This system could provide significant recovery of both energy and water. According to the literature[9], global natural gas consumption in 2018 was $3867.9 \times 10^9$ m$^3$ which corresponds to $178.7 \times 10^{12}$ mol-CH$_4$ yr$^{-1}$. With the combustion reaction $CH_4 + 2O_2 \rightarrow CO_2 + 2H_2O$, steam of a double molar amount of natural gas, is produced which indicates that the latent heat of steam is at least $15.7 \times 10^{18}$ J yr$^{-1}$. Therefore, the recovery of steam with latent heat energy would have a significant impact. In addition, the recovery of high-temperature steam ($\geq 150$ °C) further promotes energy saving due to the recovery of sensible heat together with latent heat.

For incinerator plants in Japan, water makes up approximately 50 wt% of the domestic waste feed[10], and this water becomes steam in the 900 °C combustion furnaces. In addition, combustion reactions of the waste produce additional steam (approximately 30 wt% of the waste). The output stream from combustion furnaces must be rapidly

[1]Department of Chemical Engineering, Hiroshima University, 1-4-1 Kagami-yama, Higashi-Hiroshima 739-8527, Japan. [2]PLANTEC Inc., 1-6-17 Kyomachibori, Nishi-ku, Osaka city 550-0003, Japan. [3]eSep Inc., Keihanna Open Innovation Center, 7-5-1 Seikadai, Seika-cho, Souraku-gun, Kyoto 619-0238, Japan. ✉e-mail: tsuru@hiroshima-u.ac.jp

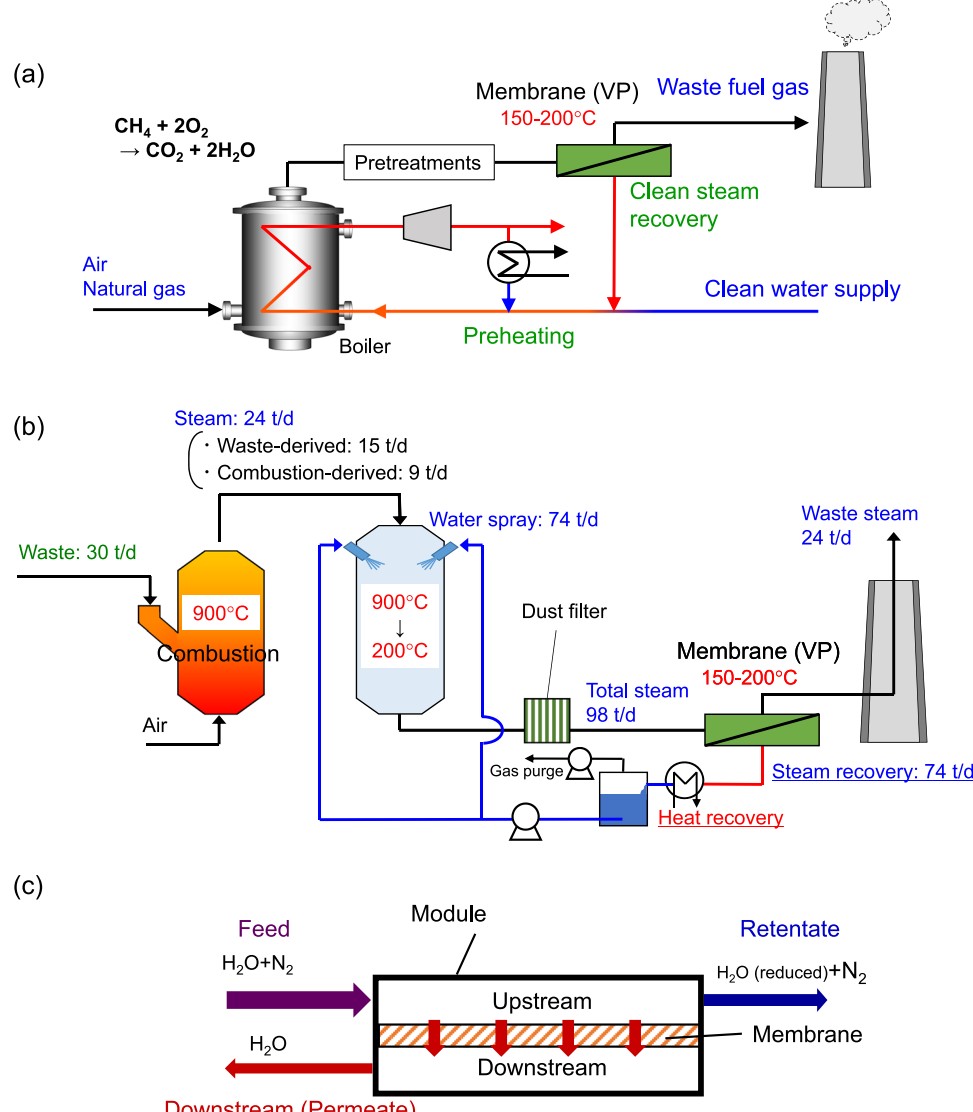

**Fig. 1 | Conceptual diagram of the proposed system. a** Schematic of steam recovery in a power plant. **b** Water balance in a self-reliant waste incinerator plant with a steam recovery membrane unit (vapor permeation: VP) (capacity: waste of 30 t day$^{-1}$). **c** A schematic membrane module that separates water vapor from the mixture of water vapor and nitrogen.

cooled to 200 °C to prevent the production of dioxin, which is produced mainly at temperatures ranging from 300 to 500 °C[11]. For this purpose, water, nearly double the amount of the waste, is often sprayed for cooling when a waste-throughput scale is <100 t day$^{-1}$[12]. Thus, the stacks release a large amount of steam that contains a significant amount of latent heat. Another negative factor is that such a large amount of steam forms what appears to be a steam condensate plume. This appearance of a steam condensate plume is a concern for the general public and often makes it difficult to construct incinerator plants in city areas. This effect is then extended to all types of industrial operations including power and chemical plants. Some plants re-heat the combustion gas stream solely to prevent the appearance of a steam condensate plume. The schematic of a waste incinerator can be found in Supplementary Note 1.

Here, we propose an environmentally friendly waste incinerator plant that uses a steam recovery membrane unit (vapor permeation, VP), as schematically shown in Fig. 1b. VP is a strategy where both the upstream and downstream of the membrane are in the vapor phase (please see Fig. 1c). These membranes have non-porous or sub-nanoporous structures that enable almost pure steam recovery in the downstream[13]. The details of the VP process can be found in

Supplementary Note 2. Herein Fig. 1b shows the proposed steam recovery system in a typical small-sized incinerator in Japan that has a waste throughput of 30 t day$^{-1}$. The stream from a combustion furnace that contains 24 t day$^{-1}$ of steam requires 74 t day$^{-1}$ of water to be sprayed on the cooling tower, so that, in total, after dust filtering the stream contains 98 t day$^{-1}$ of steam. The membrane unit allows selective steam permeation and enables steam recovery of 74 t day$^{-1}$ while maintaining the latent heat. After recovery of the latent heat using a heat exchanger, the recovered liquefied water is reused for spraying. The result is a self-reliant waste incinerator plant that does not require a supply of fresh water, which would be of great importance in any region with a shortage of water. Furthermore, when a vacuum/compressor pump was placed before the heat exchanger for the recompression of the permeating stream, the recompressed stream can be useful as a high-temperature heat medium. The schematic of this type of system can be found in Supplementary Note 1–3. Steam recovery via membranes would save energy while simultaneously recycling water. In addition, this system offers the collateral advantage of a dehumidified retentate stream from the membrane unit that requires no heating. This prevents the appearance of a steam condensate plume and results in an energy cut of 22 GJ day$^{-1}$. The advantages of this

**Table 1 | Stream properties after dust is filtered in a running waste incinerator plant in Japan with waste throughput of 30 t day⁻¹**

| Measurement month | Flow rate [m³(STP) h⁻¹] | Temperature [°C] | Composition | | | | | | | | |
|---|---|---|---|---|---|---|---|---|---|---|---|
| | | | H₂O [%] | N₂ᵃ [%] | O₂ [%] | CO₂ [%] | HCl [ppm] | NO$_x$ [ppm] | SO$_x$ [ppm] | CO [ppm] | Ash dust [g m⁻³ (STP)] |
| Feb. 2019 | 14,700 | 170 | 39.6 | 40.6 | 9.0 | 10.8 | 25 | 130 | 1 | 8 | <0.001 |
| Feb. 2019 | 13,100 | 169 | 38.1 | 42.0 | 11.4 | 8.5 | 14 | 130 | <1 | 6 | <0.001 |
| Jan. 2021 | 11,900 | 166 | 39.0 | 41.5 | 11.5 | 8.0 | 30 | 100 | 11 | 4 | <0.001 |
| Average | 13,200 | 168 | 38.9 | 41.4 | 10.6 | 9.1 | 23 | 120 | 4 | 6 | <0.001 |

ᵃCalculated via 100% minus the compositions of other products. The measurements of flow rate and temperature followed JIS Z 8808, and those of the compositions of $H_2O$, $O_2$, $CO_2$, HCl, NO$_x$, SO$_x$, CO, and ash dust followed JIS Z 8808, JIS K 0301, JIS K 0301, JIS K 0107, JIS K 0104, JIS K 0103, JIS K 0098, and JIS Z 8808, respectively[53–58]. The amount of ash dust was less than the detection limit.

process over other competing technologies are described in Supplementary Note 4.

The stream properties summarized in Table 1 show that hydrothermal, oxidizing, and acidic stability are required for VP membranes. Conventional organic membranes such as polyamide, polyimide and polyvinyl alcohol are unsuitable due to the lack of thermal stability exceeding 150 °C[14]. In addition, inorganic membranes such as LTA zeolite, and amorphous silica that were developed earlier also reportedly suffer from instability under hydrothermal and/or acidic conditions[13,15,16]. Membranes derived from SOD, FAU, MOR, MFI zeolite, metal-doped silica, and some perfluoro-polymers have been developed more recently and the selective steam permeation at high temperatures (≥ 150 °C) have been reported[17–29]. In studies reported in 2008 and 2019[30,31], however, the upper boundaries of steam permeance, that is, permeation flow rate divided by membrane area and trans-membrane pressure difference (Eqs. (1) and (2), Method section), seemed to be $10^{-6}$ mol m$^{-2}$ s$^{-1}$ Pa$^{-1}$ in this temperature range. In 2019, we found that organosilica membranes consisting of organically linked silsesquioxane structures showed excellent steam permeance of $2$–$5 \times 10^{-6}$ mol m$^{-2}$ s$^{-1}$ Pa$^{-1}$ with a steam/nitrogen permeance ratio that ranged as high as several hundred to several thousand at 150–200 °C[32]. In addition, we further improved the organosilica membranes from the viewpoint of hydrothermal stability[33]. The improved versions maintained a steam permeance of $4.7 \times 10^{-6}$ mol m$^{-2}$ s$^{-1}$ Pa$^{-1}$ and a steam/nitrogen permeance ratio of 350 during pressurized steam permeation at 200 °C for longer than two weeks, which showed they were excellent candidates for use in the steam recovery membrane unit of our proposed waste incinerator plant system.

In the present study, we propose a steam recovery system for waste incinerator plants that enables water recovery together with energy recovery. For this system, we developed organosilica membranes and conducted proof-of-concept testing in a running plant. We evaluated steam recovery from a simulated waste stream in the laboratory and from an actual waste stream in a running incinerator plant. The long-term stability of an organosilica membrane was tested for 190 days on a laboratory-scale basis. As far as we could ascertain, this is the first example of a steam recovery demonstration in a running incinerator plant. In addition, the proposed concept of steam recovery was validated by process simulation using membrane performances that had been correctly evaluated in the laboratory-scale test and using stream properties evaluated in an actual running incinerator plant.

## Results and discussion
### Membrane preparation and long-term stability testing on a laboratory scale
For the laboratory-scale tests, we prepared two organosilica membranes that are referred to here as M-1 and M-2. Figure 2a shows the molecular size dependency of single-gas permeance at 200 °C. The permeance dramatically decreased as the molecular size increased, which confirmed the effective molecular sieving of these membranes. In addition, the gas selectivities of these membranes were similar and comparable to those reported in the literature[33,34], which confirmed

the reproducibility of the membranes. For small gases such as hydrogen, both membranes recorded high permeance values of almost $10^{-6}$ mol m$^{-2}$ s$^{-1}$ Pa$^{-1}$. The molecular size of water (0.2955 nm)[35,36] approximates that of hydrogen (0.289 nm)[37], which suggests a high level of steam permeance.

Figure 2b shows the time course of steam permeance in a single system through M-1 at 150 °C. During the 9 days of the experiment, M-1 showed a high steam permeance of $6 \times 10^{-6}$ mol m$^{-2}$ s$^{-1}$ Pa$^{-1}$ with no decrease from the beginning, which confirmed the hydrothermal stability of the membranes with a bis(triethoxysilyl)ethane (BTESE)-derived organosilica separation layer and a BTESE-derived organosilica intermediate layer. After confirming the stable performance, we used the simulated waste stream for long-term steam recovery testing. Figure 2c shows the time courses for the permeance and permeance ratios of M-2 during steam recovery from the simulated waste stream. The membrane performances were periodically evaluated via $H_2O/N_2$ binary and $H_2O/N_2$/HCl ternary separation by evacuating the downstream, while simulated gas mixtures were continuously fed without evacuation. For the first 35 days, an equimolar mixture of steam and nitrogen was fed with 40 ppm of hydrogen chloride at 150 °C as the simulated waste stream. In the initial 23 days, the steam permeance and the $H_2O/N_2$ permeance ratio gradually decreased and increased, respectively, and then stabilized from day 23 onwards. Since no significant change in steam permeance was observed under single-steam permeation, as shown in Fig. 2b, the initial change in membrane performance can be ascribed to the presence of HCl. HCl reportedly catalyzes the condensation reaction of silanol groups and densifies the organosilica pore structure[38–40]. The stabilized values of steam permeance and $H_2O/N_2$ and $H_2O/HCl$ permeance ratios were evaluated on day 35, and found to be $2.0 \times 10^{-6}$ mol m$^{-2}$ s$^{-1}$ Pa$^{-1}$, 200 and 25, respectively. In the next period (Days 35–112), the air was mixed for the simulated stream instead of nitrogen, so the composition was $H_2O:N_2:O_2 = 50:40:10$ with 40 ppm of hydrogen chloride. Even under these hydrothermal, oxidative, and acidic conditions, the membrane performance remained stable. In the subsequent period (Days 112–190), the temperature of the membrane module was increased from 150 to 200 °C while maintaining the composition of the simulated stream. Of note, the membrane performances during this period were evaluated at 200 °C and are plotted as open symbols. Even at 200 °C, no significant degradation in membrane performance was observed. On day 190, the temperature was lowered to 150 °C to directly compare the last membrane performance with that of the previous period. The values for steam permeance and $H_2O/N_2$ and $H_2O/HCl$ permeance ratios at 150 °C on day 190 were $2.3 \times 10^{-6}$ mol m$^{-2}$ s$^{-1}$ Pa$^{-1}$, 170, and 27, respectively, which were almost comparable to those on day 35. The water composition in the downstream was higher than 99.3% (Supplementary Note 5). In addition, this stable steam permselectivity (high steam permeance and $H_2O/N_2$ permeance ratio) was superior to the values for other types of membranes including polymer and zeolites, as listed in the literature[17,23,24,33,36,41–44] and summarized in Supplementary Fig. 10 (Supplementary Note 5). The high steam permselectivity can be

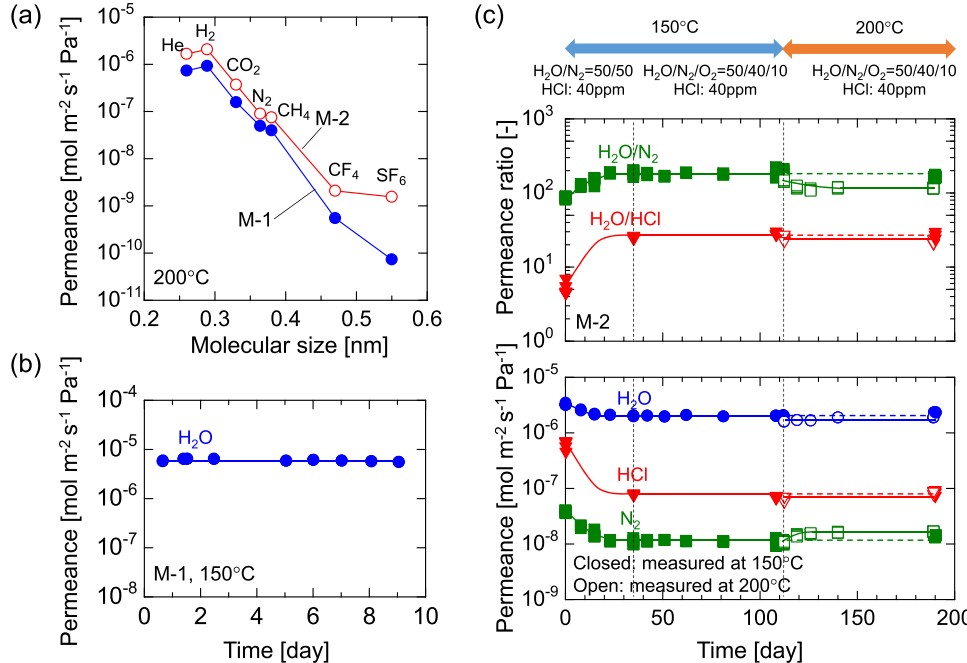

**Fig. 2 | Membrane permeation properties and durability. a** Molecular size dependency of permeating gas on single-gas permeation at 200 °C (pressure of upstream: $p_u = 200$ kPa·a, pressure of downstream: $p_d = 100$ kPa·a). **b** Time course of steam permeance in a single system during 9 days at 150 °C ($p_u = 200$ kPa·a, $p_d = 100$ kPa·a). **c** Time courses for steam, hydrogen chloride, and nitrogen permeance and their permeance ratios during 190 days at 150–200 °C ($p_u = 100$ kPa·a, $p_d = -0$ kPa·a). Source data are provided as a Source Data file.

ascribed to the molecular structure of organosilica network and the layered structure we developed, as shown in Fig. 6a. As summarized above, we confirmed the hydrothermal (~200 °C), oxidative, and acidic (HCl: ~40 ppm) stability of the organosilica membrane, as well as its steam permselectivity. This is surprising because conventional silica membranes reportedly lose their permselectivity in just 3 days under saturated water vapor at 100 °C[45]. The results indicate that organosilica membranes are quite promising for industrial use in steam recovery.

## Steam recovery performance from various types of streams

In an actual waste incinerator plant, the composition of the waste stream occasionally fluctuates depending on the type of waste and on the firing conditions. The components of most concern are acidic gases such as hydrogen chloride. Thus, we evaluated the effect of HCl concentration on membrane performance. Figure 3a shows the effect of HCl concentration on permeance and permeance ratios. Even though the membrane was exposed to an HCl concentration that reached 400 ppm at its maximum, the values for steam and nitrogen permeance evaluated at the beginning and the end of this experiment without HCl remained almost constant as shown in Supplementary Figure 11 (Supplementary Note 6). All permeances of steam, hydrogen chloride, and nitrogen, as well as all permeance ratios, were almost independent of HCl concentration. Although waste stream compositions occasionally change and HCl gas could change membrane pore structure, these results indicate the plausibility of almost continuously using organosilica membranes in a running waste incinerator plant.

Figure 3b summarizes the permeance values and permeance ratios as a function of the molecular sizes of gaseous components. $H_2O$ was the most permeable component, which indicates that the organosilica membrane could show steam-permselectivity in any separation system. The almost constant levels of steam permeance were independent of the types of mixed gases. On the other hand, the gas permeance including HCl decreased as the molecular sizes increased, which indicates the separation mechanism is based on effective molecular sieving, which is similar to single-gas permeation (Fig. 2a). As summarized in Table 1, real waste streams contain $SO_x$ and $NO_x$ on

the order of ppm, the molecular sizes of which are larger than that of oxygen[37]. Organosilica membranes would reject permeation of these large contaminants at high levels according to molecular sieving. Importantly, the permeance of steam was several times larger than that of hydrogen although the molecular size of water (0.2955 nm)[35,36] is slightly larger than that of hydrogen (0.289 nm)[37]. This cannot be explained by molecular sieving. Actually, the high levels of steam permeance were due to condensed phase diffusion, which is generally called "surface diffusion", where water molecules form a condensed phase and diffuse according to a concentration gradient across the membrane. Detailed discussions of the condensed phase diffusion phenomenon on organosilica membranes can be found elsewhere[34]. Again, the stability and molecular sieving properties can be ascribed to the structure of organosilica networks and the layered structure proposed later in Fig. 6a.

## Proof-of-concept of steam recovery in a running incinerator plant

Here, we demonstrate the proof-of-concept validation of this steam-recovery system, operated in a running waste incinerator plant over two days. Figure 4 (a) shows the time courses for the temperatures, pressures, and steam molar fractions on the first day. The membrane module was preheated to prevent any water condensation before the waste stream from the running plant was fed into it. As the experiment started, the temperatures of the feed, retentate and membrane module gradually increased and reached stable values after 1 h, while the temperature taken after the permeate had passed through the heat exchanger was cooled at 25 °C from the beginning. In addition, the pressures of the feed and retentate prior to vacuum pumping remained almost constant from the start. Of note, the relatively high pressure prior to vacuum pumping (~40 kPa·a) was caused by the insufficient evacuation capacity of the vacuum pump that was used, as explained in Supplementary Note 7. A steady operation of this steam recovery unit was realized after 1 h. However, the values for the steam molar fraction of the feed and retentate varied throughout the experiments. This was due to the changes in waste composition. The

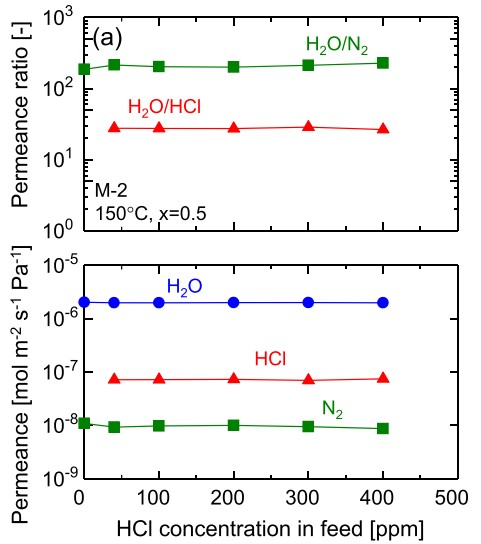

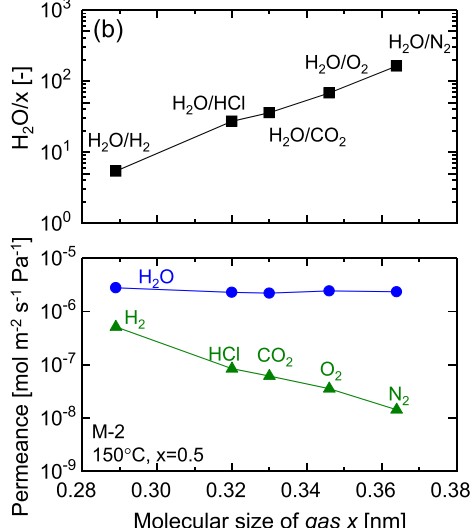

**Fig. 3 | Steam recovery from various types of gaseous fluids. a** Permeances and permeance ratios as a function of HCl concentration. This experiment was performed under $p_u$ and $p_d$ of 100 and -0 kPa-a, respectively, on day 112 (as shown in Fig. 2c). **b** Permeances and permeance ratios as a function of molecular size of non-condensable gases at 150 °C with the steam mole fraction in the feed stream was

maintained at 0.5 ($p_u$ = 100 kPa-a, $p_d$ = -0 kPa-a). $H_2O/HCl$ separation performance was evaluated using an $H_2O/N_2/HCl$ trinary system with an HCl concentration of 40 ppm in the feed, while binary systems were used to evaluate the others. This experiment was performed on days 190 and 191 (Fig. 2c). Source data are provided as a Source Data file.

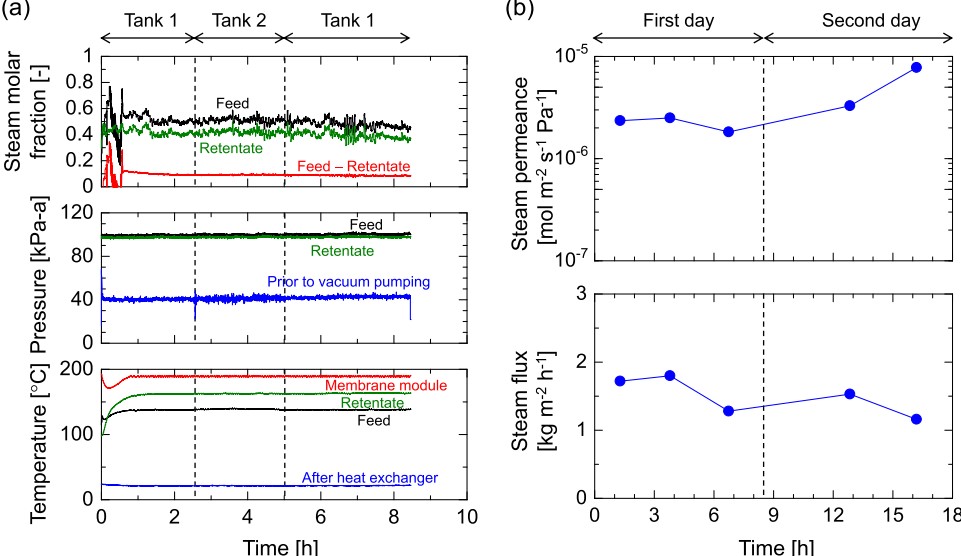

**Fig. 4 | A steam recovery demonstration in a running incinerator plant.** Time courses of **a** temperature, pressure, and steam molar fraction (the first day) and **b** steam flux and permeance. Source data are provided as a Source Data file.

steam molar fraction of the retentate was always lower than that of the feed and the difference in the molar fractions was consistent. This confirms that stable steam recovery via the membrane is independent of the waste stream composition. The recovered steam was condensed in the heat exchanger and alternately collected either into tank 1 or tank 2 to achieve the pre-determined volume. On the second day, the same experiment was repeated as the normal daily start-up and shut-down for the incinerator plant. Figure 4b shows the time course of recovered water flux and water permeance. Here, the $x$-axis indicates the effective time when the steam recovery membrane unit was operated. The plant was shut down from midnight on day 1 to the early morning of day 2. The first, second and third plots were obtained using tank 1 (first), tank 2, and tank 1 (second), respectively, on the first day. During the two days of operation, steam flux was on a similar level, although the waste stream composition changed, which again

confirmed that the steam-recovery membrane unit was successfully operated. In addition, the values for steam permeance recorded at several $10^{-6}$ mol m$^{-2}$ s$^{-1}$ Pa$^{-1}$ were almost the same as the results evaluated in the laboratory. This confirmed the applicability of organosilica membranes for steam recovery from the waste stream of an actual incinerator plant.

To further elucidate the benefits of the proposed concept of a waste incinerator with a steam recovery membrane unit, a process simulation was conducted. For this simulation, the waste stream properties in Table 1 and the membrane performance in Fig. 3b were used. Of note, the simulation ignored contaminants with a low concentration (ppm level). Figure 5 shows the effects of the membrane area on (a) steam recovery, (b) effective enthalpy recovery, (c) dimensionless steam recovery, $\theta_s$, and (d) the dew point in the retentate stream. In this figure, the solid, dashed and chain-dot

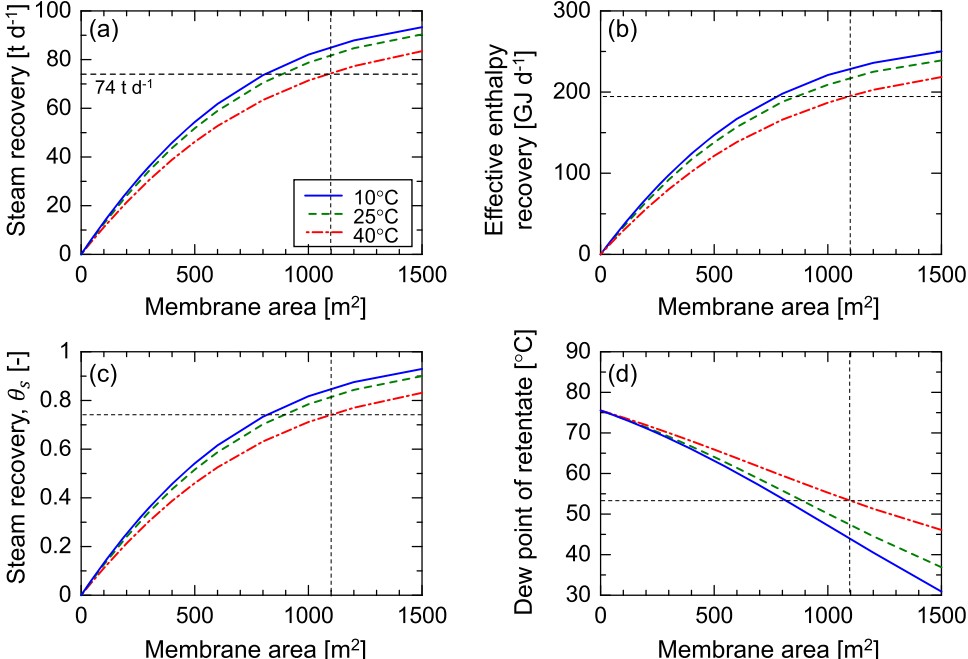

**Fig. 5 | Simulation of steam recovery performances in a 30 t d⁻¹-scale waste incinerator plant via counter-current mode.** The effects of the membrane area on **a** steam recovery, **b** effective enthalpy recovery, **c** dimensionless steam recovery, $\theta_s$, and **d** the dew point in the retentate stream. Solid, dashed and chain-dot curves indicate the temperatures of recovered water at the outlet of the heat exchanger, $T_e$, of 10, 25 and 40 °C, respectively. Average values of the feed stream (total flow rate and compositions) are summarized in Table 1, and the membrane performances appearing in Fig. 3b were assumed.

curves represent the temperatures of the recovered water at the outlet of the heat exchanger, $T_e$, which are 10, 25, and 40 °C, respectively. In Fig. 5a, the steam recovery increases as the membrane area increases and as $T_e$ decreases. $T_e$ approximated the temperature of the coolant at the inlet of the counter-current heat exchanger, so that a value for $T_e$ that is higher than the environmental temperature would be useful for regular operation. In the case of $T_e = 40$ °C, a membrane area of 1100 m² achieved steam recovery of 74 t day⁻¹, which equals the water spray rate in a cooling tower, and achieves the proposed self-reliant status for a waste incinerator plant that could operate without a water supply. In addition, in emergencies such as a disaster, the water supply could be stopped. Such a self-reliant waste incinerator plant could recover an additional amount of clean water, nearly 10 t day⁻¹, by reducing the $T_e$ from 40 to 10 °C. This recovered water could potentially supply drinking water to 6700 people, following additional conventional purification treatments such as filtration and adsorption. Figure 5b shows that the effective enthalpy recovery, estimated according to Eq. (5), followed a trend similar to that of the steam recovery. This is because the recompression energy for permeating non-condensable gases such as $N_2$ was very small due to the high level of steam-permselectivity of the membrane. With a membrane area of 1100 m² and a $T_e$ of 40 °C, it would be possible to recover energy as high as nearly 200 GJ/day. This represents approximately 70% of the waste combustion heat[10], confirming the concept of energy recovery. Details of the available energy such as the temperature and the flow rate of coolant as well as the recovery and operational energy can be found in Supplementary Notes 3-4 and 3-5. A membrane area of 1100 m² and $T_e$ would enable the recovery of 74 t day⁻¹ of steam, which corresponds to a 74% recovery of the steam from the waste stream, as shown in Fig. 5c. Therefore, in Fig. 5d, the dew point of the retentate stream could be reduced from 76 to 53 °C without a decrease in the waste stream temperature of about 170 °C because the permeation of steam occurs isothermally. This low dew point would prevent the formation of a steam condensate plume from the stacks without the need for post-heating.

We proposed a concept for an environmentally friendly waste incinerator plant that uses organosilica membranes consisting of developed molecular networks and layered structures to accomplish steam recovery and confirmed its effectiveness. The present study will be applicable not only to waste incineration plants but also to other types of industrial operations such as power plants and chemical plants. This would have a great impact on industries from environmental and public-relations perspectives.

## Methods

### Membrane fabrication

The organosilica membranes were fabricated via the sol-gel method following the procedure[46]. Bis(triethoxysilyl)ethane (BTESE, 97% purity) purchased from Gelest Inc. was mixed with ultrapure water and hydrochloric acid (35 wt% of aqueous solution) in ethanol solvent (99.5 wt% purity). After stirring at 50 °C for 1 h, BTESE-acid sol was prepared via hydrolysis and condensation of BTESE. To prepare the BTESE-swing sol, an aqueous solution of ammonia (28%) was added to the BTESE-acid sol at an $NH_3$/HCl molar ratio of 6 to promote further condensation. Then, after stirring at room temperature (RT) for 40 min, hydrochloric acid was added with a molar ratio of HCl/$NH_3 = 0.8$ to return the pH to an acidic state. This procedure was used to terminate the condensation reaction and obtain a stable BTESE-swing sol. The initial molar ratio of BTESE/$H_2O$/HCl was $1/240/10^{-1}$–$10^{-2}$ while the concentration of BTESE was 5 wt%.

Figure 6a features a schematic diagram of an organosilica membrane composed of a support, an intermediate layer, and a separation layer; the fabrication procedure was first established elsewhere[33]. As a support, an α-alumina porous tube (average pore size: 1 μm, Nikkato Corp., Japan) was used. First, the concentration of the α–alumina particles (diameter: 1–2 and 0.2 μm) was adjusted to 5–10 wt% with BTESE-swing sols, and then they were coated on the outside surface of the support for smoothing. This procedure was followed by heat treatment at 300 °C under a $N_2$ flow. Then, BTESE-swing sols were coated onto the smoothed surface, and heat treatment was conducted at 300 °C under a $N_2$ flow to form a BTESE-derived intermediate layer

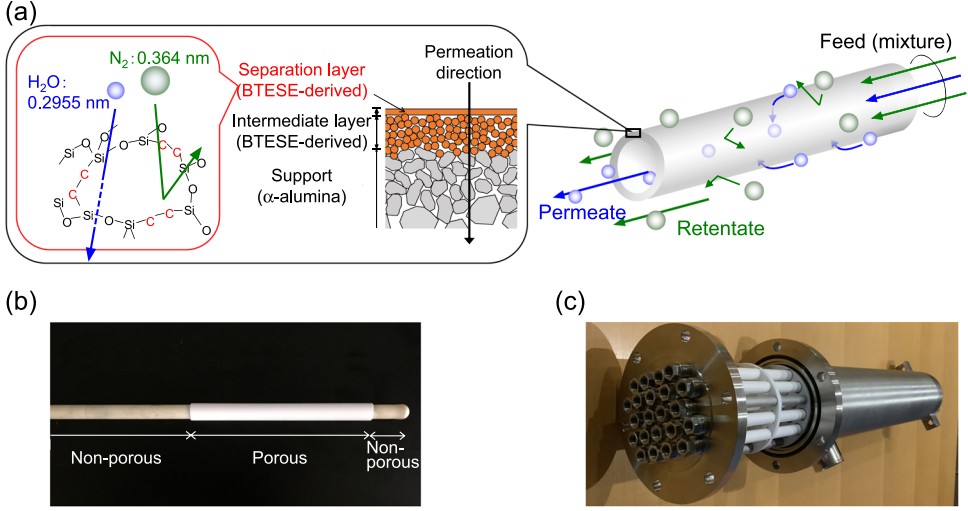

**Fig. 6 | Membranes used in this study. a** Schematic of a BTESE-derived organo-silica membrane. **b** A membrane (developed by Hiroshima University) used for laboratory-scale tests. **c** Membranes and the membrane module (produced by eSep, Inc.) that were used for the bench-scale test (steam-recovery demonstration in a running waste incinerator plant).

with a pore size of about 1 nm. After that, a BTESE-derived separation layer was formed by coating a BTESE-acid sol onto the intermediate layer, which was followed by heat treatment at 300 °C under an $N_2$ flow. The BTESE-derived network pores in the separation layer enable selective steam permeation via molecular sieving[47]. A cross-sectional scanning electron microscope (SEM) image of the membrane can be found in Supplementary Note 8, and the chemical structure of the organosilica separation layer has been characterized elsewhere[33,47,48]. Fig. 6b features a picture of the BTESE-derived membrane with a membrane area of $3.1 \times 10^{-3}$ m$^2$ that was used for laboratory-scale tests. The white portion is the porous membrane, the right and left sides of which were plugged and connected to an α−alumina nonporous tube, respectively. Steam selectively permeates from outside to inside through the porous membrane, then the permeant flows from right to left. Figure 6c shows the BTESE-derived membranes and the membrane module that was used for a bench-scale test, which served as a steam recovery demonstration for a running incinerator plant. The membrane module contained 22 membranes with 12 of them plugged for the demonstration to allow adjustments to the permeating flow rate. Thus, the effective membrane area was 0.12 m$^2$.

## Single-component permeation experiment

Single-component permeation (steam, He (99.995%), $H_2$ (99.99%), $CO_2$ (99.5%), $N_2$ (99.995%), $CH_4$ (99.99%), $CF_4$ (99.999%), and $SF_6$ (99.9%)) was evaluated on a laboratory-scale using the in-house-fabricated apparatus described in our previous papers[33]. The single components were fed to the outside of the membrane at 200 kPa-a, while the inside of the membrane was maintained at 100 kPa-a. For single-steam permeation, liquid ultrapure water was completely evaporated at -250 °C prior to being fed into the membrane module to prevent any water condensation. The permeate flow rates of gases were measured using a film-flow meter, and that of steam was measured by trapping permeated steam (downstream) in a cold trap during a pre-determined period of time. The permeance of single-component permeation was calculated via Eq. (1).

$$\Pi = \frac{Q}{A(P_u - P_d)} \tag{1}$$

In Eq. (1), $\Pi$, $Q$, $A$, $P_u$, and $P_d$ are the permeance [mol m$^{-2}$ s$^{-1}$ Pa$^{-1}$], the permeating flow rate [mol s$^{-1}$], the membrane area [m$^2$], and the upstream and downstream pressures [Pa], respectively.

## Steam recovery testing in the laboratory and in a running incinerator plant

Steam recovery testing from simulated waste gases was performed on a laboratory scale using the in-house-fabricated apparatus illustrated in Fig. 7a. Since the simulated waste steam contains steam, nitrogen, oxygen, and hydrogen chloride, the evaporator, the membrane module, and the tubes between them were made of glass to prevent corrosion. The controlled flow rates of the HCl aqueous solution and gases were mixed, which was followed by evaporation before entering the membrane module. The permeating flow rate of steam was evaluated from its trapped weight during a pre-determined period of time, and that of the non-trapped gases was evaluated by measuring the flow rate after the operation of the vacuum pump. The flow rates of HCl were evaluated according to the trapped weight and the electrical conductivity of the trapped solution. For the cold trap, liquid nitrogen (−196 °C) was used when hydrogen and nitrogen were fed as non-condensable gases, while a mixture of dry ice and methanol (−79 °C) was used when carbon dioxide, oxygen and methane were fed. The permeance in steam recovery was calculated using Eq. (2).

$$\Pi_i = \frac{Q_i}{A \frac{(p_{u,i} - p_{d,i})_1 - (p_{u,i} - p_{d,i})_2}{\ln \frac{(p_{u,i} - p_{d,i})_1}{(p_{u,i} - p_{d,i})_2}}} = \frac{Q_i}{A(p_{u,i} - p_{d,i})_{lm}} \tag{2}$$

In Eq. (2), $\Pi_i$, $Q_i$, $p_{u,i}$, and $p_{d,i}$ are the permeance [mol m$^{-2}$ s$^{-1}$ Pa$^{-1}$], the permeating flow rate [mol s$^{-1}$], and the partial pressure both upstream and downstream [Pa], respectively, of the $i$th component. The subscripts 1 and 2 indicate the inlet and outlet of the membrane module, respectively. This equation has been commonly used, even though its use contains some limitations of the operating conditions. In the present work, the operation conditions of steam recovery followed previously reported guidelines[49].

Steam recovery from an actual waste stream was tested in a running waste incinerator plant using the bench-scale apparatus illustrated in Fig. 7b. A blower was placed in the retentate stream to draw the actual waste stream into the membrane module from a bypass line of the main duct and discharge the retentate stream back to the bypass line. The properties of the actual waste gases appear in Table 1. The downstream was cooled via a heat exchanger and liquefied water was trapped in the tank until a pre-determined volume of water was reached. Tanks-1 and -2 were alternatively used for the continuous measurement. The permeating non-condensable gases were

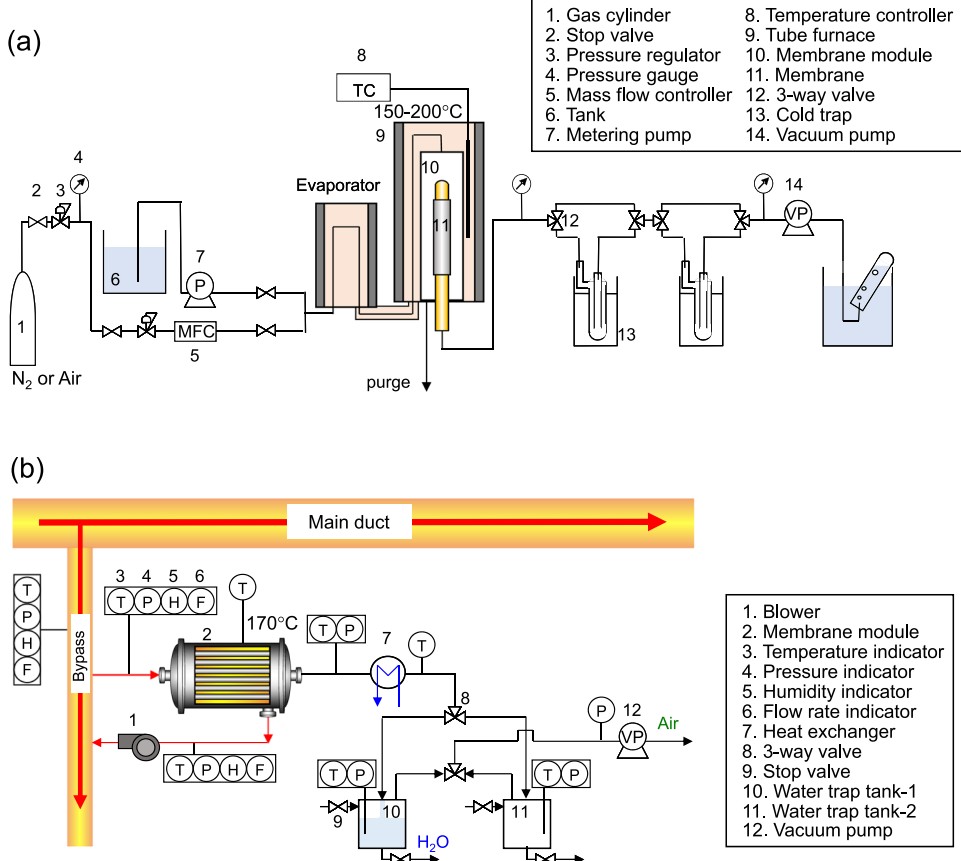

**Fig. 7 | Schematics of the equipment for steam recovery testing. a** Steam recovery from a simulated waste stream on a laboratory-scale. **b** Steam recovery from an actual waste stream in a running incinerator plant (bench-scale).

evacuated to maintain an invacuo downstream. The permeating flow rate and permeance of the steam were calculated from the amount of time required to trap the pre-determined volume of water using Eq. (2).

**Process simulation to validate the concept**

Numerical simulation of quaternary separation ($H_2O$, $N_2$, $O_2$, $CO_2$) was performed in a counter-current mode to evaluate the effectiveness of the steam recovery membrane unit in a running waste incinerator plant (waste: 30 t day$^{-1}$). The details of this type of simulation are schematically illustrated in Supplementary Note 3-2 and example calculations are shown in Supplementary Note-3-3 to 3-5. The accuracy of the model has been confirmed elsewhere[49–52]. Under the assumption of a plug flow, the mass balance of the $i$th component in the upstream is expressed by Eq. (3).

$$-\frac{\mathrm{d}F_i}{\mathrm{d}A} = \Pi_i\left(p_{\mathrm{u},i} - p_{\mathrm{d},i}\right) \quad (3)$$

In Eq. (3), $F_i$, $p_{\mathrm{u},i}$, and $p_{\mathrm{d},i}$ indicate the molar flow rate in the upstream, and the partial pressures in both the upstream and downstream of the $i$th component, respectively, all of which are variable depending on the membrane area, $A$. $\Pi_i$ is the permeance of the $i$th component, which could be evaluated via the steam recovery experiment.

The mass balance of the $i$th component in the downstream is expressed by Eq. (4).

$$\frac{\mathrm{d}Q_i}{\mathrm{d}A} = \Pi_i\left(p_{\mathrm{u},i} - p_{\mathrm{d},i}\right) \quad (4)$$

In Eq. (4), $Q_i$ is the molar flow rate of the $i$th component in downstream.

The molar flow rate of each component in the upstream and in downstream along the membrane was integrated under the initial conditions, as summarized in Table 1.

The effective enthalpy recovery via the membrane unit was calculated according to Eq. (5).

$$\text{Effective enthalpy recovery} = L\upsilon_{T_e}Q_s + \sum_{i=1}^{4}\left(\int_{T_e}^{T_p} Q_i c_{\mathrm{p,g}}\mathrm{d}T\right) - \int_{p_{s,T_e}}^{P_{\mathrm{atm}}} \frac{Q_n R T_e}{P}\mathrm{d}P \quad (5)$$

In Eq. (5), $Q_s$ [mol s$^{-1}$], $Q_n$ [mol s$^{-1}$], and $Q_i$ [mol s$^{-1}$] indicate the permeating flow rates of steam, non-condensable gases and the $i$th components, respectively. $T_p$ [K] and $T_e$ [K] are the temperatures of the process waste stream and the permeate stream following the use of a heat exchanger, respectively. $L\upsilon_{T_e}$ [J mol$^{-1}$], $c_{\mathrm{p,g}}$ [J mol$^{-1}$ K$^{-1}$], $R$ [J mol$^{-1}$ K$^{-1}$], and $P$ [Pa] are the latent heat of water at $T_e$, the heat capacity of gas, the gas constant, and the pressure, respectively. The temperature dependency of the heat capacity $c_{\mathrm{p,g}}$ can be found in Supplementary Note 9. On the right side of this equation, the first, the second and the third terms indicate the recovered latent heat of steam, the recovered sensible heat of the permeating components via a heat exchanger, and the isothermal compressing energy used for the vacuum pump to evacuate the permeating non-condensable gases, respectively. The dew point of the retentate stream was calculated using the Antoine equation, as described in Supplementary Note 9.

## Scanning electron microscopy

The cross-sectional morphology of the organosilica membrane was examined using a field emission scanning electron microscope (FE-SEM, Hitachi S-4800, Japan) at an applied voltage of 3.0 kV. To mitigate charging, the sample was coated with palladium.

## Data availability

All experimental data generated or analyzed during this study are included in this published article, its supplementary information and Source Data files. Source data are provided with this paper.

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

## Acknowledgements

This research was supported by the Environment Research and Technology Development Fund (JPMEERF20201G02 (T.T.)) of the Environmental Restoration and Conservation Agency provided by Ministry of the Environment of Japan, Grants-in-Aid for Scientific Research KAKENHI (19K22085 (T.T.), 20H05227 (T.T.), 22H04551 (T.T.), 22K18922 (T.T.)), and JSPS Grants-in-Aid for JSPS Fellows Number 19J22400 (N.M.).

## Author contributions

N.M.: Investigation (laboratory-scale experiments & simulation), writing-original draft. A.T.: Investigation (bench-scale test), writing-review and editing. T.Y.: Investigation (bench-scale test), writing-review and editing. K.S.: Development of membranes and membrane module for bench-scale, writing-review and editing. K.G.: Development of membranes and membrane module for bench-scale, writing-review and editing. H.N.: Writing-review and editing. M.K.: Writing-review and editing. T.T.: Conceptualization, supervision, writing-review and editing

## Competing interests

The authors declare no competing interests.
