## [Peer Review File · Nature Communications]

Steam recovery from flue gas by organosilica membranes for simultaneous harvesting of water and energyEditorial Note: This manuscript has been previously reviewed at another journal that is not operating a transparent peer review scheme. This document only contains reviewer comments and rebuttal letters for versions considered at *Nature Communications*. Mentions of prior referee reports and the other journal have been redacted.

REVIEWER COMMENTS

Reviewer #1 (Remarks to the Author):

[redacted]

With the exception of the comments below, I believe the authors have addressed my earlier comments.

1. I appreciate the additional details regarding Cases I and II of the steam recovery systems. The equation used to calculate the compression energy of the vacuum pump for both cases and any assumptions involved (e.g., compression efficiency, adiabatic vs. isentropic compression) should be provided.
2. There seems to be a semantics issue regarding the description of how feed is delivered to the bench-scale membrane unit (as in Figure 7b). The use of the word “introduce” implies to me (and I think many readers) that the blower is upstream of the membrane. Instead of the current wording “A blower was used to introduce the actual waste stream from a bypass line of a duct to the membrane module and to push out the retentate stream back to the bypass line,” I think a more accurate statement would be “A blower was placed in the retentate stream to draw the actual waste stream into the membrane module from a bypass line of the main duct and discharge the retentate stream back to the bypass line.”

[redacted]

Reviewer #2 (Remarks to the Author):

This manuscript reported a steam recovery demonstration in a running incinerator plant which enables simultaneous water and energy recovery. Organosilica membranes were developed for this system and the long-term stability was tested. This work is interesting. [redacted]. This work can be accepted after clarifying the novelty in the introduction part.

[redacted]

Response to reviewer's comments

For reviewer-1

The authors thank reviewer-1 for his/her review. Following are the responses (in black) to each comment (in red), and the revised parts of the manuscript are shown in blue.

Comment-1-0

[redacted]

With the exception of the comments below, I believe the authors have addressed my earlier comments.

Response-1-0

[redacted]

[redacted]

Please see the following responses to other comments.

Comment-1-1

I appreciate the additional details regarding Cases I and II of the steam recovery systems. The equation used to calculate the compression energy of the vacuum pump for both cases and any assumptions involved (e.g., compression efficiency, adiabatic vs. isentropic compression) should be provided.

Response-1-1

Thanks for reviewer-1's feedback. We are glad that reviewer-1 showed agreement with the proposed steam recovery system. The following is our answer for the additional comment.

In the manuscript, the compression energy was calculated as $\int_{p_s, T_e}^{P_{atm}} \frac{Q_n R T_e}{P} dP$. This is the third term on Eq. (5) in the main manuscript. In the caption of Fig. S8, we wrote just as "The energies were estimated according to Eq. (5) in the main manuscript." We have revised the manuscript to emphasize the equation.

The equation assumes isothermal compression with ideal compression efficiency. From the viewpoint of energy recovery, adiabatic and isentropic compressions are desirable because isothermal compression releases the compression heat to the environment, indicating energy loss. However, we selected isothermal compression because adiabatic and isentropic compressions increase the temperature of recovered stream to high temperature where a pump cannot work. Following is an example calculation.

Eq. (R1) expresses the temperature at the outlet of an adiabatic compression pump [Chemical engineering handbook (Kagaku Kogaku Binran), The society of Chemical Engineers, Japan (2011)].

$$T_{out} = T_{in} \left[\left(\frac{p_{out}}{p_{in}} \right)^{\gamma-1/\gamma\eta} - 1 \right] + T_{in} \quad (R1)$$

T , p , γ , and η indicate temperature, pressure, adiabatic index ($=c_p/c_v$), and compression efficiency, respectively. The subscripts of *in* and *out* indicate the positions at the inlet and outlet of the pump, respectively.

For an example of Case II, $T_{in}=441$ K, $p_{in}=7.384$ kPa-a, and $p_{out}=101.3$ kPa-a. Since the stream fed to the pump is almost pure steam (ref. Table S7), γ can be assumed as 1.39 (γ of water vapor at 168°C [NIST chemistry WebBook, Thermophysical Properties of Fluid System, Standard reference database number 69]). Under assumption of $\eta=0.9$, $T_{out}=996$ K is estimated. In addition, in the same assumptions, $T_{out}=641$ K is estimated for Case I. These high temperature streams are attractive,

however, no pumps work.

Again, we assumed isothermal compression which decrease energy efficiency due to energy loss. However, the steam recovery system still shows effective heat recovery as shown in Fig. S8 (Supporting Information).

In order to clarify our assumptions for the calculation clear, the manuscript has been revised as follows.

(Page 28, Line 18)

In Eq. (5), Q_s [mol/s], Q_n [mol/s], and Q_i [mol/s] indicate the permeating flow rates of steam, non-condensable gases and the i -th components, respectively. T_p [K] and T_e [K] are the temperatures of the process waste stream and the permeate stream following the use of a heat exchanger, respectively. Lv_{T_e} [J/(mol)], $c_{p,g}$ [J/(mol K)], R [J/(mol K)], and P [Pa] are the latent heat of water at T_e , the heat capacity of gas, the gas constant, and the pressure, respectively. The temperature dependency of the heat capacity $c_{p,g}$ can be found in SI-9 (Supporting Information). On the right side of this equation, the first, the second and the third terms indicate the recovered latent heat of steam, the recovered sensible heat of the permeating components via heat exchanger, and the isothermal compressing energy used for the vacuum pump to evacuate the permeating non-condensable gases, respectively. The dew point of the retentate stream was calculated using the Antoine equation, as described in SI-9 (Supporting Information).

(Supporting Information, Page 25, Line 2)

Fig. S8 shows the quantities of recovered steam, along with the recovered and consumed energy, based on the above simulation. Additionally, the cost of the recovered steam and energy consumed by the vacuum pump were calculated based on the cost to produce steam²² and the cost of green electricity²³, since the heat recovery and energy consumed by the vacuum pump cannot be directly compared. Here, we assumed the construction of the membrane unit as summarized in Tables S5 (for Case I) and S6 (for Case II). The compression energy for vacuum pumps was calculated via Eq. (S19) under assumption of isothermal compression with ideal compression efficiency.

$$W = \int_{P_{in}}^{P_{out}} \frac{QRT}{P} dP \quad (S19)$$

In Eq. (S19), W , P_{out} , P_{in} , Q , and T are energy consumption, pressures at the outlet and the inlet of the pump, molar flow rate, and temperature, respectively.

Comment-1-2

There seems to be a semantics issue regarding the description of how feed is delivered to the bench-scale membrane unit (as in Figure 7b). The use of the word “introduce” implies to me (and I think many readers) that the blower is upstream of the membrane. Instead of the current wording “A blower was used to introduce the actual waste stream from a bypass line of a duct to the membrane module and to push out the retentate stream back to the bypass line,” I think a more accurate statement would be “A blower was placed in the retentate stream to draw the actual waste stream into the membrane module from a bypass line of the main duct and discharge the retentate stream back to the bypass line.”

Response-1-2

We thank reviewer-1 for providing accurate description. The manuscript has been revised following this.

(Page 26, Line 10)

Steam recovery from an actual waste stream was tested in a running waste incinerator plant using the bench-scale apparatus illustrated in Fig. 7 (b). A blower was placed in the retentate stream to draw the actual waste stream into the membrane module from a bypass line of the main duct and discharge the retentate stream back to the bypass line. The properties of the actual waste gases appear in Table 1.

Comment-1-3

Comment retraction: I thank the authors for their response regarding former equation S15 and my comment about constant heat capacity. As the authors noted, constant heat capacity is not inherently assumed in the equation (since it is located inside the integral). I cannot recreate my reasoning behind my comment. I apologize for expending the authors’ time in responding to that comment.

Response-1-3

We appreciate the reviewer's profound comments, which we believe have improved the present manuscript.

For reviewer-2

The authors thank reviewer-2 for his/her review. Following are the responses (in black) to each comment (in red), and the revised parts of the manuscript are shown in blue.

Comment-2-0

This manuscript reported a steam recovery demonstration in a running incinerator plant which enables simultaneous water and energy recovery. Organosilica membranes were developed for this system and the long-term stability was tested. This work is interesting.

[redacted]

Response-2-0

The authors would like to thank Reviewer-2 for showing interest in our work and positive comments to the present manuscript.

[redacted]

REVIEWERS' COMMENTS

Reviewer #1 (Remarks to the Author):

I am satisfied with the revisions made by the authors and believe the manuscript is acceptable for publication.

Reviewer #2 (Remarks to the Author):

My earlier comments have been addressed.